# The Impact of Telemedicine on Patients with Hidradenitis Suppurativa in the COVID-19 Era

**DOI:** 10.3390/healthcare11101453

**Published:** 2023-05-17

**Authors:** Marcin Gierek, Diana Kitala, Wojciech Łabuś, Justyna Glik, Karol Szyluk, Kornelia Pietrauszka, Beata Bergler-Czop, Paweł Niemiec

**Affiliations:** 1Dr Sakiel Center for Burns Treatment, Jana Pawła II Street 2, 41-100 Siemianowice Śląskie, Poland; 2Department of Organisation of Chronic Wound Healing, Faculty of Health Sciences in Katowice, Medical University of Silesia in Katowice, 40-055 Katowice, Poland; 3Department of Physiotherapy, Faculty of Health Sciences in Katowice, Medical University of Silesia in Katowice, 40-752 Katowice, Poland; 4District Hospital of Orthopaedics and Trauma Surgery, Bytomska 62 Street, 41-940 Piekary Slaskie, Poland; 5Department of Dermatology, Medical University of Silesia in Katowice, Francuska Street, 40-027 Katowice, Poland; 6Department of Biochemistry and Medical Genetics, School of Health Sciences in Katowice, Medical University of Silesia in Katowice, Medykow Street 18, 40-752 Katowice, Poland

**Keywords:** hidradenitis suppurativa, acne inversa, telemedicine, teleconsultation, teledermatology, telehealth, COVID-19, healthcare

## Abstract

(1) Background: Hidradenitis suppurativa is a chronic, inflammatory skin disease. It is characterized by the transformation of normal skin into skin with abscesses, nodules, tunnels, and scars. The most commonly affected areas are the armpits, groins, buttocks, and subscapular area. Patients with HS require constant care under the supervision of the outpatient clinic. Due to the COVID-19 pandemic, consultations have been introduced in the form of telemedicine. The aim of this study was to evaluate the availability of HS treatment during the COVID-19 pandemic and to assess patient satisfaction, problems with access to medical care, and the impact of the pandemic on the course of the disease. (2) Methods: An internet survey with an anonymous questionnaire was used to assess the effectiveness of telemedicine consultations. The survey consisted of 25 closed questions, and responses were kept fully anonymous. (3) Results: Most respondents reported minor problems with accessing specialized HS medical care during the COVID-19 pandemic (n = 25, 35.71%). However, 35.71% (n = 25) of them reported major problems with appointments for specialized ambulatory treatment during the last few months of the pandemic, mainly due to delayed appointments. Almost half of the respondents had been diagnosed with COVID-19 (n = 34, 48.57%), and 58.57% (n = 41) of respondents did not see a correlation between COVID-19 infection and HS progression. (4) Conclusions: Our study showed that the pandemic significantly limited access to medical advice, and patients with hidradenitis suppurativa prefer standard consultations.

## 1. Introduction

A new viral disease, COVID-19, was first identified in late 2019 in Wuhan, China. The World Health Organization (WHO) declared that there was a pandemic on 11 March 2020 [1]. As of the 5th of January 2023, the WHO COVID-19 situation report indicated that there have been over 761 million confirmed cases and more than 6.8 million deaths across all continents [2].

A significant number of worldwide deaths have been reported due to the COVID-19 pandemic, resulting in difficulties in providing the continuity of care and globally maintaining the capacity of health systems [3,4,5,6]. The global situation, which initially assumed that the pandemic would be halted, has placed an increased emphasis on telemedicine as a means of providing remote medical care across all medical specialties [7]. With the growth in the number of Internet users and advancements in technology, the availability of telemedicine services has become increasingly popular. In 2018, the global telemedicine market was valued at USD 38 million. By 2024, the market for telemedicine services is projected to exceed USD 103 million [8]. However, problems with the availability of services, including access and implementation of telemedicine, remain a global issue [9].

Despite historical reports in the literature linking physicians with distant patients [10,11], telemedicine has become the standard for consultation during a pandemic.

Prior to the pandemic, telemedicine consultations were not commonly practiced by physicians [12]. The terms telehealth and telemedicine are often used interchangeably, but the U.S. Health and Services Administration defines telehealth as a broader term that encompasses a wider range of technologies used for providing healthcare and health-related education at a distance [13]. The Centers for Medicaid and Medicare Services define telemedicine as “two-way, real-time, interactive communication between a patient and a physician or practitioner in a remote location” [14]. Telemedicine should be evaluated by physicians on a case-by-case basis, taking concerns such as avoiding unlicensed practice across jurisdictions [15], adherence to ethical aspects of telemedicine [16], telemedicine-specific legal considerations [17], and other limitations on telemedicine [18] into consideration. In addition, physicians often cite a lack of training, high equipment costs, and increased liability as major challenges to implementing telemedicine [19].

Hidradenitis suppurativa (HS) is a chronic and highly painful inflammatory skin disease with a complex etiology involving multiple factors. The affected skin develops abscesses, inflamed nodules, tunnels, and eventually, scars. The most commonly affected areas are the armpits, groin, buttocks, and subscapular region [20]. Due to persistent symptoms such as pain, the discharge of pus, a foul odor, and accompanying itchiness, HS can have a significant negative impact on the health-related quality of life of patients. Patients with HS are more prone to depression and often experience social stigmatization [20]. The treatment for HS involves dermatological, and sometimes, surgical intervention to alleviate the pain caused by abscesses.

Due to the potential risk of contagion, dermatologic consultations have been postponed in many countries [21]. However, consultations for outpatients receiving biological treatments were still conducted in order to maintain the continuity of care [22]. Urgent consultations were the only ones that were allowed.

During the COVID-19 pandemic, providing daily telemedicine visits for patients with hidradenitis suppurativa (HS) has raised concerns about the effectiveness of medical services. Patients have expressed dissatisfaction with this form of medical advice, and the ability to effectively treat and control their symptoms has been limited. Therefore, the aim of this study is to evaluate the availability of HS treatment during the COVID-19 pandemic and assess patients’ satisfaction with the care they have received. This study also aimed to evaluate the challenges in accessing medical care and to assess the impact of the COVID-19 pandemic on the management of HS.

## 2. Materials and Methods

This was a retrospective cohort study conducted using an anonymous survey questionnaire. The survey comprised 25 close-ended questions and was designed to maintain the full anonymity of participants.

This was an original questionnaire developed based on the analysis of issues reported by patients who contacted our Center for Burns Treatment in Siemianowice Śląskie, Poland. The questionnaire addressed health concerns, local deterioration, and difficulties in accessing primary medical care during the COVID-19 pandemic. It should be noted that the unique needs of HS patients and the specificities of our healthcare system prompted us to create our own original questionnaire to evaluate the quality of medical services during the pandemic. The questionnaire is available in the Appendix A.

This study utilized a survey with three parts. The first part included six questions related to demographics and education level. The second part focused on HS, including the time of the onset of the first symptoms, time from the onset of the first symptoms to diagnosis, clinical phenotype, pain (assessed on a 10-point Visual Analog Scale), and independence in changing dressings (eight questions in total). The third part evaluated the impact of the COVID-19 pandemic on specialist treatment of HS, including the availability of care, impact on health, dates of consultations, teleconsultations, and the assessment of the effectiveness of teletherapy in treating HS (11 questions). Both printed and electronic versions of the survey were used, with the latter ones disseminated via an internet forum for people diagnosed with HS. The study was conducted over two months from 1 February 2022 to 31 March 2022, and data were managed using Excel. The study was ethically conducted in compliance with the World Medical Association Declaration of Helsinki, and as it used fully anonymous survey data, it did not require Bioethics Committee approval under Polish law (the study used fully anonymous survey data).

### Statistical Analysis

All statistical analyses were conducted using Statistica, ver. 12 software (StatSoft, Dell Inc., Round Rock, TX, USA). The normality of data distribution was assessed using the Shapiro–Wilk test, and homogeneity of variances was evaluated using the Levene test. The Mann–Whitney U test was used to compare two independent samples for statistical hypothesis testing, while the Kruskal–Wallis test was employed to compare more than two groups of independent samples that did not meet the normality assumption. For dichotomous data, the chi-square test was used. The significance level was set at 0.05 (5%).

## 3. Results

Table 1 presents basic data on the respondents, including demographic information and level of education. A total of 70 respondents completed the questionnaire, with the majority of them being male (n = 54; 77.14%). The analysis of the respondents’ level of education revealed that most underwent higher education (n = 32; 45.71%), while only 5.71% declared undergoing medical education. Additionally, most of the respondents lived in cities with a population of around 50,000 or more than 200,000 inhabitants (equally n = 23; 32.86%) (Table 1).

The majority of respondents reported experiencing the first symptoms of the disease more than 10 years ago (n = 32; 45.71%). About 57.14% (n = 40) of the respondents were officially diagnosed with HS 1–5 years ago. Despite most of the respondents having appointments with dermatologists or surgeons for more than a year (n = 41; 58.57%), the majority (n = 57; 81.43%) believed that the disease was diagnosed too late. Around 65.71% (n = 46) of the respondents reported experiencing daily pain sensation, with 17.14% (n = 12) of them having reported a VAS score of four. For most patients, changing dressings is uncomfortable but not painful (48.57%, n = 34), and 77.14% (n = 54) of respondents reported being able to change their dressings daily alone or with the help of a family member (Table 2).

### 3.1. Legend: VAS, Visual Analogue Scale

Most of the respondents (n = 25; 35.71%) reported minor problems accessing specialist medical care for hidradenitis suppurativa (HS) during the COVID-19 pandemic. However, 25 respondents (35.71%) reported major problems with specialist appointments in the last few months, mainly due to appointment delays. Almost half of the respondents (n = 34; 48.57%) reported being diagnosed with COVID-19, and 41 respondents (58.57%) did not see a correlation between COVID-19 infection and HS progression. Half of the patients (n = 35; 50.00%) reported never having had their medical appointment postponed or cancelled due to the pandemic. The most common waiting time for medical appointments was 7–14 days (n = 21; 30.00%). However, during the pandemic, the waiting time for specialist medical appointments was mostly over two months (n = 24; 34.29%). Almost half of the respondents (n = 32; 45.71%) reported never having attempted to schedule a tele-appointment. Of the respondents, 54 (77.14%) declared that the quality of telemedicine appointments was not sufficient, and 59 (84.29%) reported that it is not possible to treat HS this way (Table 3).

### 3.2. Legend: HS, Hidradenitis Suppurativa

There was a significant difference between the age of the respondents and the time that passed from the onset of the first symptoms (*p* = 0.006). Differences were present between participants aged 21–30 and 41–50 years old (*p* = 0.005) and between those who were 31–40 and 41–50 years old (*p* = 0.018). Sex, place of residence, and level or type of education did not have an impact on the time when the first symptoms occurred. The same results were obtained when the impact on the time from the onset of the first symptoms to the final diagnosis was assessed; only the age of the participants had an impact (*p* = 0.010), but there was statistically significant difference between the of 16–20 and 41–50 years olds (*p* = 0.037) and between 21–30 and 41–50 years olds (*p* = 0.024). The only parameter that had an impact on the opinion that the disease was diagnosed too late was the time from the onset of the first symptoms to diagnosis (*p* = 0.001). The regularity of medical appointments due to symptoms was associated with the period of time when the first symptoms began (*p* = 0.0026) and the time of the final diagnosis (*p* = 0.012), but also with the level of education (*p* < 0.001). Post hoc test revealed that there was a difference between the group with a time of the onset of the first symptoms lasting 5–10 years and over 10 years (*p* = 0.035). There were also differences between patients diagnosed 1–5 years after the first symptoms and >10 years in the regularity of appointments (*p* = 0.025) and with patients with basic and vocational levels of education (*p* < 0.001).

Surprisingly, there was a difference in the daily sensation of pain and the place of living (*p* = 0.019; Figure 1), education level (*p* = 0.042), and the opinion held about the disease being diagnosed too late (*p* = 0.013). However, it was not confirmed by the patients’ reported VAS values.

None of the sociodemographic parameters had an impact on the sensation during dressing changes. However, sex had an impact on the ability of the patient or their family member to change dressings (*p* = 0.025). Over 83% of men were able to change their dressings, while only 56.30% of women were able to do so. There was a difference between the daily pain sensation and opinion on problems with appointments in ambulatory care (*p* = 0.012).

Patients who reported being able to perform self-care during dressing changes (or receiving care from a family member) had lower VAS scores than those who reported being unable to perform self-care did (median ± QD: 4.00 ± 2.50 vs. 7.50 ± 1.50, respectively). Additionally, patients diagnosed with COVID-19 had lower VAS scores than those who did not report having the disease did (median ± QD: 4.00 ± 2.00 vs. 6.00 ± 2.00, respectively). Pain during dressing changes had a significant impact on the patients’ opinions about there being new problems with appointments with professionals providing ambulatory care (*p* = 0.005). There were differences between different levels of daily pain and the patients’ opinions on the worsening of HS symptoms during the pandemic (*p* = 0.036). Furthermore, there were differences in VAS scores between the patients’ answers to the question about the severity of their HS symptoms during the pandemic (*p* = 0.003). Patients who reported no changes in their HS symptoms had the lowest VAS scores, while those who reported critical deterioration had the highest VAS scores (Figure 2). Only the difference between patients who reported no changes and those who reported major deterioration was statistically significant (*p* = 0.006). None of the parameters had an impact on waiting times before or during the pandemic. Age was the only parameter that significantly influenced the use of telemedicine (*p* = 0.012).

None of the tested factors had an impact on the opinion about the quality of telemedicine appointments. Only daily pain sensation was a parameter that significantly influenced people’s opinions about the effectiveness of telemedicine in the treatment of HS (*p* = 0.028). In Figure 3, we present various aspects of telemedicine consultations. The majority of respondents answered that HS cannot be treated via telemedicine consultations.

## 4. Discussion

In the presented study, the majority of survey participants had contact with telemedicine. A total of 2.86% of survey participants reported using telemedicine very often, 7.14% reported that they frequently use it, and 44.29% reported that they rarely use it. Among the patients with hidradenitis suppurativa, as many as 22.86% reported having great difficulty changing dressings on their own. Additionally, 27.14% of the patients experienced enormous difficulties in obtaining medical care during the COVID-19 pandemic. Minor and major problems were reported by a total of 48.57% of patients in the study. Prior to the pandemic, the most common waiting time for an appointment was 7–14 days (30.0% of patients). During the pandemic, the most frequently selected answer by respondents for the average waiting time was >2 months (34.19% of patients in the survey). Furthermore, 77.14% of respondents reported that telemedicine care is insufficient, and most of the respondents were dissatisfied with telemedicine consultations.

In a randomized trial conducted by Armstrong et al. [23], which involved 296 participants, the adjusted difference between the online and in-person consultation groups in the mean change in self-reported PASI score over the 12-month study period was −0.27 (95% CI, −0.85 to 0.31). Armstrong et al. concluded that the online collaborative model was as effective as standard consultations are in improving the outcomes for dermatological patients. They also suggested that modern telehealth service delivery models emphasize collaboration, quality, and efficiency, which could be helpful for improving patient-centered outcomes in chronic diseases, including skin diseases [23].

The results of a study by Villa et al. [24] indicate that the use of telemedicine in dermatological emergencies is safe and effective. Villa et al. argue that the telemedicine assessment of dermatological diseases is a useful and practical alternative in the clinical care of emergency patients [24].

According to the results presented by Armstrong et al. [25], the remote model was found to be as effective as the care provided during a standard clinic consultation is [25]. In a study by Rios et al. [26], unblinded correlation analysis was conducted in which dermatologists were randomly allocated to two groups: standardized examinations and telemedicine examinations. Both groups evaluated 30 cases and assessed the degree of correlation (Cohen’s κ coefficient) between the diagnoses made by each group [26]. In the study group, 63.3% were female, and 30% of the patients were aged between 50 and 59 years. A good correlation (κ = 0.6512) was found between the telemedicine and face-to-face examination results. The study did not find significant differences in the diagnostic skills of the two groups. Rios et al. concluded that teledermatology can be effectively used to facilitate diagnosis during patient-specific teleconferences with patients who cannot attend in person, but face-to-face consultations remain the gold standard [27].

Domogolla et al. [27] conducted a study to investigate the impact of an eHealth Smartphone App on the mental health of patients with psoriasis. The study included 107 patients and was randomized. There were 53 patients in the control group and 54 in the intervention group. The study lasted for 60 weeks, and 71.9% of patients completed it. The authors found a significant reduction in depression (HADS-D) in the intervention group. The results showed that the significant improvement in HADS-D score depended on the frequency with which patients used the app. Based on the study’s results, Domogolla et al. [27] conclude that using a smartphone app for disease management can be an important tool to achieve long-term improvements in mental health in dermatological patients. However, they also recommend further research to analyze the newly observed impact of the frequency of app use [27].

In a study conducted by Armstrong et al. [28], the effectiveness of a face-to-face online model was evaluated for follow-up dermatological care among pediatric and adult patients with atopic dermatitis and compared to that of in-person standard clinic visits. A total of 156 children and adults were enrolled in this randomized trial. The percentages of patients who were cured or nearly cured of the disease (IGA score 0 or 1) were 38.4% (95% CI, 27.7% to 49.3%) in the face-to-face online group and 43.6% (95% CI, 32.6% to 54.6%) in the in-person group (standard outpatient clinic consultations). The difference in the proportion of patients who were cured or nearly cured between the two groups was 5.1%, which fell within the 10% equivalence margin previously established by the authors [28]. The authors concluded that the direct online access model produces equivalent improvements in clinical outcomes for atopic dermatitis patients to those of in-person care. Telemedicine visits may represent a novel type of dermatology service delivery for patients with chronic skin diseases [28].

In a study by Zelickson et al. [29], a model of teledermatology in a nursing home was presented. In this study, a nurse collected and transmitted histories and images using a telemedicine system, and diagnosis and treatment plans were determined based on the analysis of uploaded images and patients’ history. An independent dermatologist performed an on-site dermatology consultation within 2 days after the data were uploaded to the system. The study included 29 patients with 30 skin conditions, and correct diagnoses were made in 60 (67%) of 90, 51 (85%) of 60, and 53 (88%) of 60 patients. A correct treatment plan was included in 63 (70%) of 90, 52 (87%) of 60, and 54 (90%) of 60 patients. Dermatologists found telemedicine treatment and diagnosis to be convenient. The study presented by Zelickson et al. provides evidence that teledermatology consultations in nursing homes can replace some on-site consultations by offering high-quality care via teledermatology [29].

In a study by Ruggiero et al. [30], the use of teledermatology for the treatment of hidradenitis suppurativa was evaluated. The authors found that patients generally preferred face-to-face visits, particularly for areas affected by HS that were considered to be sensitive. Patients also expressed concerns about sharing video or photo documentation, feeling that it was not safe to do so. These findings support the idea that in-person consultations may be preferred over teledermatology in some cases. However, in another study by Ruggiero et al. [31], teledermatology was found to play a significant role during the COVID-19 pandemic, as it provided a means for remote consultation and treatment. Both patients and physicians reported high satisfaction rates with the use of teledermatology. The authors acknowledged, however, that there are still concerns about privacy and medico-legal issues, as well as a lack of defined and secure online teledermatology platforms. In a study by Marasca et al. [32], it was found that skin diseases worsened during the COVID-19 pandemic, likely due to decreased access to in-person care and delayed treatment. These findings highlight the need for continued development and implementation of remote healthcare technologies, such as teledermatology, to ensure that patients have access to high-quality care even during times of crisis. Hidradenitis suppurativa (HS) is a medical condition characterized by deep-seated nodules, which require physical examination, and this cannot be substituted with image evaluations. An ultrasound is a more precise tool than palpation is, and it is commonly used to monitor patients during treatment due to its non-invasive nature [33]. Okeke et al. suggest that virtual consent forms and telemedicine platforms with online imaging may be potential solutions for pandemic-related consultations. However, the authors acknowledge that challenges, such as patient privacy, comfort, difficulty visualizing anatomical areas, and the inability to palpate the patient, still need to be addressed [34]. Several studies have reported that teleconsultation was the primary method of consultation for HS during the pandemic, and telemedicine played a leading role [35,36,37,38,39,40]. However, the use of teledermatology for HS management has not been widely investigated, and fewer studies are available. Patel et al. showed that face-to-face visits are more important for HS management than tele-visits are. Since HS is an unstable disease and often involves intimate body areas, the use of video or photographic assessments should be handled with caution. Patients with HS also have a high prevalence of depression and anxiety [38]. In their study, 41 patients undergoing 73 remote consultations were compared with 40 subjects attending 70 face-to-face visits [38].

A survey conducted by Kang et al. on Facebook that supported patient groups with HS reported that subjects with severe HS disagreed that teledermatology provided equally effective care compared with that of face-to-face visits [39]. The authors recommended that patients send photos of private areas instead prior to their virtual appointment, but patients may feel reluctant and uncomfortable with exposing intertriginous (especially genital) areas via videoconferencing. However, the study design of Kang et al. was a Facebook questionnaire, and their results are in contrast with our study. Our study showed that patients with HS preferred standard consultations, but further research is needed in this area [39]. In conclusion, HS management requires physical examination and ultrasound monitoring, which cannot be replaced with teledermatology. While telemedicine has played a leading role in HS management during the pandemic, challenges such as patient privacy, comfort, and the inability to palpate the patient remain. Therefore, face-to-face consultations are still important for HS management, and the use of teledermatology should be handled with caution. Future research is necessary to fully evaluate the effectiveness of teledermatology for HS management.

However, intelligent computing models are used for forecasting the outbreak of COVID-19. Maybe these could help to predict pandemics in the future [40].

The COVID-19 pandemic posed a significant global challenge for healthcare systems. The widespread adoption of telemedicine as a standard for conducting medical consultations during the pandemic has highlighted the need for further research and improvements in the field. The systematization of telemedicine services can lead to improvements in the management of chronic skin diseases such as hidradenitis suppurativa. Future research should be conducted to identify the best practices for delivering telemedicine services for patients with HS, taking into consideration patient privacy, comfort, and the need for physical examinations in certain cases.

## 5. Conclusions

Telemedicine emerged as a necessary and widely used method of medical consultation during the pandemic. However, our study findings indicate that this approach may not be effective for patients with hidradenitis suppurativa who require standard consultations, which is the preferred method of consultation for this patient population. Our study also highlighted the significant limitations in access to medical advice during the pandemic. Therefore, future research should be conducted to improve telemedicine platforms, as it is unknown whether we will face similar challenges in the future.

## Figures and Tables

**Figure 1 healthcare-11-01453-f001:**
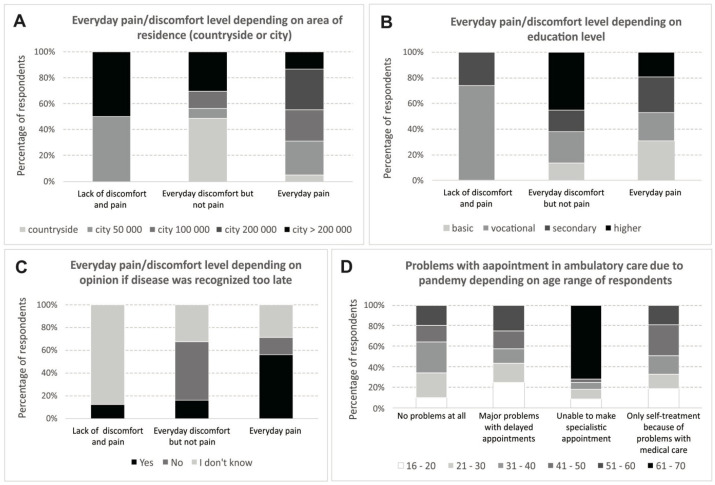
Pain complaints depending on selected factors: (**A**) area of residence, (**B**) educational level, (**C**) opinion on whether HS (Hidradenitis suppurativa) was recognized too late, and (**D**) age of patients.

**Figure 2 healthcare-11-01453-f002:**
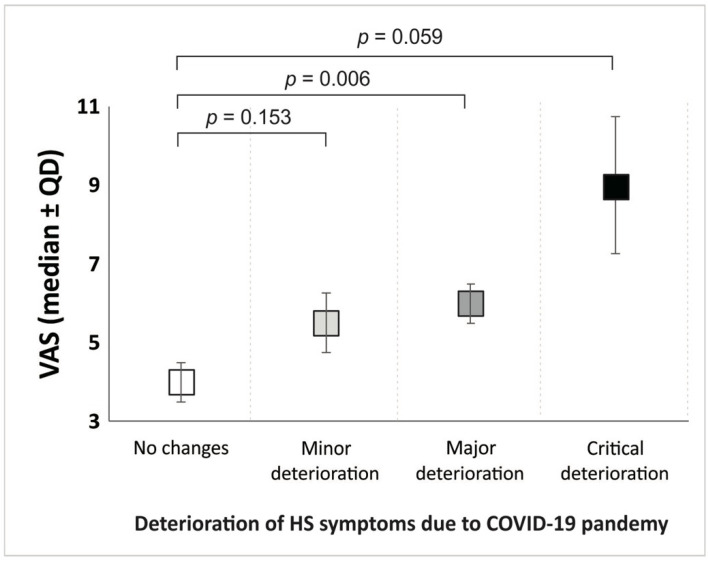
Deterioration of HS symptoms in COVID-19 pandemic in relation to the VAS scale.

**Figure 3 healthcare-11-01453-f003:**
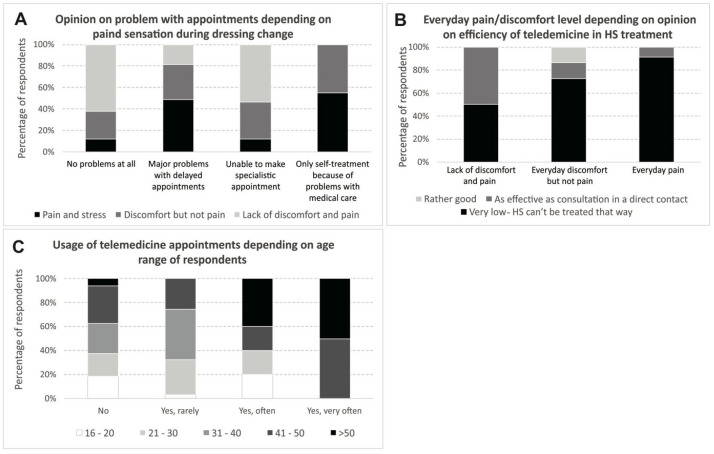
Different aspects of telemedicine consultations. (**A**) Opinion about appointment problems according to pain experienced during dressing changes. (**B**) Daily pain/discomfort depending on opinion about effectiveness of telemedicine in HS treatment. (**C**) Use of telemedicine visits according to age of respondents.

**Table 1 healthcare-11-01453-t001:** Characteristics of survey participants.

Demographic Parameter		n (%)
Age range (years)	16–20	8 (11.43)
	21–30	16 (22.86)
	31–40	21 (30.00)
	41–50	20 (28.57)
	51–60	4 (5.71)
	61–70	1 (1.43)
	<70	0 (0)
Sex	female	16 (22.86)
	male	54 (77.14)
Place of residence	countryside	6 (8.57)
	city with about 50,000 inhabitants	23 (32.86)
	city with about 100,000 inhabitants	9 (12.86)
	city with about 200,000 inhabitants	9 (12.86)
	city with above 200,000 inhabitants	23 (32.86)
Education level	basic	7 (10.00)
	secondary	23 (32.86)
	higher	32 (45.71)
	vocational	8 (11.43)
Type of education	medical	4 (5.71)
	non-medical	66 (94.29)
Hospitalized in Burn Treatment Center	yes	16 (22.86)
	no	54 (77.14)

**Table 2 healthcare-11-01453-t002:** Characteristics of survey participants diagnosis and daily experiences.

Disease Experience Parameter		n (%)
Time from fist symptoms (years from now)	<1	2 (2.86)
	1–5	11 (15.71)
	5–10	25 (35.71)
	>10	32 (45.71)
Time to official diagnosis (years from now)	<1	18 (25.71)
	1–5	40 (57.14)
	5–10	2 (2.86)
	>10	10 (14.29)
Was disease diagnosed too late?	yes	57 (81.43)
	no	5 (7.14)
	I do not know	8 (11.43)
Time of regular medical appointments due to symptoms (months)	1	2 (2.86)
	1–6	4 (5.71)
	6–12	23 (32.86)
	>12	41 (58.57)
Daily pain sensation	Every day pain	6 (8.57)
	Everyday discomfort but nor pain	23 (32.86)
	Lack of permanent discomfort and pain	9 (12.86)
Daily pain level (VAS)	0	3 (4.29)
	1	7 (10.0)
	2	4 (5.71)
	3	5 (7.14)
	4	12 (17.14)
	5	7 (10.0)
	6	7 (10.0)
	7	8 (11.43)
	8	9 (12.86)
	9	5 (7.14)
	10	3 (4.29)
Sensation during dressing change	Pain and stress	25 (35.71)
	Discomfort but not pain	34 (48.57)
	Lack of discomfort and pain	11 (15.71)
Ability to take selfcare during dressing change (or care by family member)	Yes	54 (77.14)
	No	16 (22.86)

**Table 3 healthcare-11-01453-t003:** Characteristics of appointment issues in patients with HS.

Appointment or COVID-19 Issues		n (%)
Problems with medical care regarding HS during pandemic	No differences	10 (14.29)
	Minor problems	25 (35.71)
	Major problems	15 (21.43)
	Enormous problems	19 (27.14)
	Unable to make appointment at all	1 (1.43)
Lately problems with appointment in ambulatory care	No problems at all	18 (25.71)
	Major problems with delayed appointments	25 (35.71)
	Unable to make specialistic appointment	7 (10.00)
	Only self-treatment because of problems with medical care	19 (27.14)
Deterioration of symptoms due to problems with appointments during pandemic	No changes	21 (30.00)
	Minor deterioration	30 (42.86)
	Major deterioration	13 (18.57)
	Critical deterioration	6 (8.57)
Diagnosis of COVID-19 infection	yes	34 (48.57)
	no	36 (51.43)
Impact of COVID-19 infection on worsening of HS symptoms	Yes, definitely worse	7 (10.00)
	No impact	22 (31.43)
	There is no connection between HS and COVID	41 (58.57)
Medical appointment delayed or cancelled because of COVID-19 pandemic	Never	35 (50.00)
	Very rarely	13 (18.57)
	Often	9 (12.86)
	Very often	9 (12.86)
	All the time	4 (5.71)
Time of waiting for specialistic medical appointment before pandemic	Almost immediately	9 (12.86)
	7–14 days	21 (30.00)
	14–21 days	20 (28.57)
	21 days–2 months	9 (12.86)
	>2 months	11 (15.71)
Time of waiting for specialistic medical appointment during pandemic	Almost immediately	4 (5.71)
	7–14 days	10 (14.29)
	14–21 days	17 (24.29)
	21 days–2 months	15 (21.43)
	>2 months	24 (34.29)
Usage of telemedicine appointments	No	32 (45.71)
	Yes, rarely	31 (44.29)
	Yes, often	5 (7.14)
	Yes, very often	2 (2.86)
Quality of telemedicine appointments	Normal and personal	10 (14.29)
	Insufficient	54 (77.14)
	Rather good	6 (8.57)
	Very good	0 (0.0)
Efficiency of treatment HS by tele-appointments	Very low—HS cannot be treat that way	59 (84.29)
	Rather good	3 (4.29)
	It can be as efficient as personal treatment is	8 (11.43)
	Very good way of treatment HS	0 (0.00)

## Data Availability

Not applicable. Data is not available due to the anonymity of the patients and other ethical reasons.

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
