# Peer review of "The Impact of Telemedicine on Patients with Hidradenitis Suppurativa in the COVID-19 Era"

_healthcare, 2023, doi:10.3390/healthcare11101453_

Round 1

Reviewer 1 Report

The research is more like a telephonic survey which may not be consider as original research.

1) the methodology is very limited.

2) No original research contribution.

3) Reference 20 to 24 self citing and many others is observed while reviewing the paper.

4) 25 closed questions are not sufficient and not comprehensive.

5) Not a novel research.

Author Response

Many thanks for the invaluable reviewer work with this manuscript and many thanks for all the feedback.

We have tried to improve the manuscript in accordance with the reviewers' recommendations. Thank you very much for any feedback and comments that will significantly improve the paper.  All new parts of the paper are highlighted in red.

1) We have improved and expanded the methodology sections to make the role of this paper more specific.

2) We have included our own questionnaire that was used in the study (Supplementary material).

3) We have expanded the references sections to include new items relevant to the topic. That's right, we have also cited our papers - the main author deals with the surgical treatment of inverted acne - all our papers are up to date.

4) The aim of the questionnaire (our own questionnaire) was to assess primarily the problems that patients faced in our country.  We found significantly longer appointments for which patients were waiting, patients outside (HS patients) were not satisfied with these types of appointments.

5) Thank you for your feedback, but this is an original work with our own original questionnaire based primarily on the evaluation of specific HS patients in our country. Perhaps the reviewer's assessment is due to the differences between different health systems - it is difficult to assess the system of UK, USA or Asian countries or our country (Poland). Each health system has its own peculiarities and significant difference - for this reason we also used our own questionnaire.

We thank you for all your suggestions - we hope that in its present form the manuscript is more readable and satisfactory to the Honourable Reviewer.

Best regards,

Marcin Gierek

MD, PhD.

Reviewer 2 Report

Include a list of four to ten key words after the Abstract
The aim is not clear in the abstract section
Mention major contribution and paper organization clearly
More recent literature survey required, and compare with them
Improve the results and discussion section, add more sentences for proper justifying the works
Try to improve the usage of English grammar. The formatting, grammar and typo errors should be carefully checked before processing this article.
The introduction of the article is very concise and short. From the Introduction section 1, it is rather unclear what gaps the manuscript attempts to explore and the overall added value of the paper. The contribution of the article should be highlighted clearly in the introduction section along with organization of paper.
Discussions and comparative results with non-ML approaches need to be provided;

Consider related works, Intelligent computing on time-series data analysis and prediction of COVID-19 pandemics, COVID-19 diagnosis system by deep learning approaches

Improves the quality of the figure

Mention the features distributions
The complexity analysis is required;

A serious proofreading of the manuscript is required.

Author Response

Many thanks for the invaluable reviewer work with this manuscript and many thanks for all the feedback.

We have tried to improve the manuscript in accordance with the reviewers' recommendations. Thank you very much for any feedback and comments that will significantly improve the paper.  All new parts of the paper are highlighted in red.

  • We add more key words as suggested
  • We try to specify aim of the study in the Abstract and in the Introduction section more clearly
  • As suggested we add new references
  • We changed the discussion section and add new references which will improve the quality of this paper
  • Manuscript had extensive English editing with native.
  • We add new sentences in Introduction section to specify the aim and to present the manuscript more clearly – we also add a questionnaire in the Supplementary Material.
  • Discussion was changed according to your suggestions.

Thank you very much for all efforts with this manuscript.

Best regards,

Marcin Gierek

MD, PhD.

Reviewer 3 Report

Taking into account that telemedicine is a very vast field, it would be useful to frame the intervention of teleconsultation in this subject in a systematic way, which are involved and what is analysed. How the intervention was designed.

If telemedicine is well guided by ethical and legal principles and by the data protection policy. the principles of patient privacy are guaranteed.

In subjects and methods:

what is the intervention design

What data were collected from the questionnaires to assess the effectiveness of telemedicine?

It is not clear .

only satisfaction? and disease-directed treatment?

What prevents the effectiveness of the treatment, the degree of patient satisfaction?

 Moderate editing of English language

Author Response

Many thanks for the invaluable reviewer work with this manuscript and many thanks for all the feedback.

We have tried to improve the manuscript in accordance with the reviewers' recommendations. Thank you very much for any feedback and comments that will significantly improve the paper.  All new parts of the paper are highlighted in red.

1) Yes, telemedicine is a very broad topic, but because of the differences in healthcare systems, we focused on the problems and complaints about teleconsultation that patients reported to us. We designed our own questionnaire-which is in Supplementary Material.

2) In our opinion, it is very difficult to manage HS patients via teleconsultation - due to the lack of physical examination facilities, however, there are authors who believe that it is possible to manage such patients despite these challenges. The lead author is a surgeon and treats surgical so IHS4 /Hurley III patients and has the most contact with such patients.  Such patients are difficult to manage by teleconsultation - which of course is possible but difficult.

3) The questionnaire we used is available to view in Supplementary Material.

4) We have improved the number of references and added new items, which makes the text clearer.

5) We have made the Material and Methods more detailed and extensive.

6) The text has been proofread by a Native Speaker.

Thank you very much for all efforts with this manuscript. It will improve the quality of this manuscript.

Best regards,

Marcin Gierek

MD, PhD.

Reviewer 4 Report

Dear Authors,

herein the corrections I suggest

ABSTRACT "Hidradenitis suppurativa is a chronic , inflammatory skin disease"... add characterized by deep seated nodules, abscesses and fistulas in the apocrine bearing areas of the body, such as...

please delete "subscapular"

ABSTRACT, last line two typos to be corrected "PATIENTS with hidradenitis suppurativa PREFER"

INTRODUCTION. Delete this "A very high number of deaths worldwide were reported"

INTRODUCTION. Please add: "Since dermatologic consultations demonstrated to be a potential cause of contagion [Kwatra SG, et al. Dermatology practices as vectors for COVID-19 transmission: a call for immediate cessation of non-emergent dermatology visits. J Am Acad Dermatol. 2020; 82(5): e179– e180.], in many countries  the outpatient dermatological consultations were postponed, while consultations for outpatients treated with biologicals were confirmed in order to guarantee therapeutic continuity [Nazzaro G, et al. What is the role of a dermatologist in the battle against COVID-19? The experience from a hospital on the frontline in Milan. Int J Dermatol. 2020 Jul;59(7):e238-e239. doi: 10.1111/ijd.14926.]. Only urgent consultations were admitted.

INTRODUCTION This sentence is not complete "Despite historical reports in the literature linking physicians with distant patients [10,11]"

Also this sentence "In HS, normal skin transforms into 67 skin with the formation of abscesses, inflamed nodules, tunnels, and finally scars. The 68 most frequently affected areas are the armpits, groin, buttocks and subscapularis" should be modified according to the abstract. Moreover [20-24] are all citations from the same group. I suggest to add also other authors, such as "Zouboulis CC, et al. What causes hidradenitis suppurativa ?-15 years after. Exp Dermatol. 2020 Dec;29(12):1154-1170. doi: 10.1111/exd.14214. PMID: 33058306."

In the discussion you should reduce the paragraphs dedicated to psoriasis. HS is a disease characterized by deep seated noduled where the physical examination is mandatory and cannot be substituted by images. Furthermore, you did not mention the role of ultrasound in hidradenitis that can modify the clinical scores in a number of patients [Nazzaro G, et al. Comparison of clinical and sonographic scores in a cohort of 140 patients with hidradenitis suppurativa from an Italian referral centre: a retrospective observational study. Eur J Dermatol. 2018 Dec 1;28(6):845-847. doi: 10.1684/ejd.2018.3430.]. Ultrasound is more precise than palpation and is used to monitor patients during treatment thanks to non invasive tools.

I agree with your conclusions.

-

Author Response

Many thanks for the invaluable reviewer work with this manuscript and many thanks for all the feedback.

We have tried to improve the manuscript in accordance with the reviewers' recommendations. Thank you very much for any feedback and comments that will significantly improve the paper.  All new parts of the paper are highlighted in red.

  • All manuscript has been proofread by a Native Speaker.
  • We add all suggested references by the Reviewer.
  • We add some new parts in the text

Thank you very much for your kind review. It will improve the quality of this manuscript.

Best regards,

Marcin Gierek

MD, PhD.

Round 2

Reviewer 1 Report

The followings are my observations and suggestions while reviewing the paper.

1) This is more like a survey rather than a research focusing on COVID-19. Telephonic survey can be used as a secondary tool but it is highly recommended to go for Diagnostic Biochemistry and Clinical Medicine procedure along with Computational Image analysis with supervised professional in these field. 

2) It Could be more convincing if there are research review with proper analysis.

3) Many self-cited papers, paper number 22, 23, 24 etc. should not be included as per the recommendations of this journal. Please remove any self cited journals.

Author Response

1) Indeed - it was not a research focusing on COVID-19. Yes Reviewer is right this was internet survey concerning the attitude of polish society with HS for telemedicine - and that was the aim of the study, which was in the text in many parts. (we were not measuring the effects of Covid-19 on patients) Only 1 question in our questionnaire was about general opinion about health after covid. you can broader check our questionnaire which is in Supplementary Material. This was not a phone call survey it was anonymous internet survey. 

2) Thank you for your suggestion, manuscript was changed due to your valuable comments

3) Thank you very much - we remove our self- citations. However, our scientific work is focused on HS, and we do not think that this citations were wrong or do anything improper in this manuscript( main author have multiple articles about HS). We removed it with your suggestion, but we leave also this matter to the main Scientific Editor. 

Thank you very much for your comments, we hope that our explanation will satisfies you. 

Reviewer 2 Report

Authors addressed well

Improvements require as per journal standards

Author Response

We would like to thank once again for your invaluable work with our manuscript. 

We hope that present form of the manuscript enroll all your important suggestions.

Thank you very much.

Sincerely yours,

Marcin Gierek

Reviewer 3 Report

I believe the manuscript has been
sufficiently improved to warrant publication in Healthcare.

I believe the manuscript has been
sufficiently improved to warrant publication in Healthcare.

Author Response

(The authors gave the same response as above.)

Reviewer 4 Report

The paper has been improved after this revision 

Author Response

(The authors gave the same response as above.)
